# Nontoxic and Naturally Occurring Active Compounds as Potential Inhibitors of Biological Targets in *Liriomyza trifolii*

**DOI:** 10.3390/ijms232112791

**Published:** 2022-10-24

**Authors:** Israa M. Shamkh, Mohammed Al-Majidi, Ahmed Hassen Shntaif, Peter Tan Deng Kai, Ngoc Nh-Pham, Ishrat Rahman, Dalia Hamza, Mohammad Shahbaz Khan, Maii S. Elsharayidi, Eman T. Salah, Abdullah Haikal, Modupe Akintomiwa Omoniyi, Mahmoud A. Abdalrahman, Tomasz M. Karpinski

**Affiliations:** 1Botany and Microbiology Department, Faculty of Science, Cairo University, Cairo 12613, Egypt; 2Chemo and Bioinformatics Lab, Bio Search Research Institution, BSRI, Giza 12613, Egypt; 3Department of Chemistry, College of Science for Women, University of Babylon, Alhilla 51002, Iraq; 4Victoria Junior College, Crimson Research Institute, Singapore 449035, Singapore; 5Department of Biotechnology, Faculty of Biology and Biotechnology, VNU-HCM University of Science, Ho Chi Minh City 700000, Vietnam; 6Department of Basic Dental Sciences, College of Dentistry, Princess Nourah bint Abdulrahman University, Riyadh 11564, Saudi Arabia; 7Zoonoses Department, Faculty of Veterinary Medicine, Cairo University, Giza 11221, Egypt; 8Children’s National Hospital, Washington, DC 20010, USA; 9Central Public Health Laboratories, Egyptian Ministry of Health, Cairo 11511, Egypt; 10Biochemistry Department, Faculty of Science, Ain Shams University, Ain Shams 11591, Egypt; 11Department of Pharmacognosy, Faculty of Pharmacy, Mansoura University, Mansoura 35516, Egypt; 12Department of Chemistry, Faculty of Science, University of Lagos, Lagos 101017, Nigeria; 13Science Department—Chemistry, Milton Academy, Crimson Research Institute, Milton, MA 02186, USA; 14Chair and Department of Medical Microbiology, Poznań University of Medical Sciences, Wieniawskiego 3, 61-712 Poznań, Poland

**Keywords:** protein inhibitors, *Liriomyza trifolii*, molecular docking, inhibitory activity, protein-ligand interactions, yeast extraction, bean-leaf extraction

## Abstract

In recent years, novel strategies to control insects have been based on protease inhibitors (PIs). In this regard, molecular docking and molecular dynamics simulations have been extensively used to investigate insect gut proteases and the interactions of PIs for the development of resistance against insects. We, herein, report an in silico study of (disodium 5′-inosinate and petunidin 3-glucoside), (calcium 5′-guanylate and chlorogenic acid), chlorogenic acid alone, (kaempferol-3,7-di-O-glucoside with hyperoside and delphinidin 3-glucoside), and (myricetin 3′-glucoside and hyperoside) as potential inhibitors of acetylcholinesterase receptors, actin, α-tubulin, arginine kinase, and histone receptor III subtypes, respectively. The study demonstrated that the inhibitors are capable of forming stable complexes with the corresponding proteins while also showing great potential for inhibitory activity in the proposed protein-inhibitor combinations.

## 1. Introduction

A concerning problem that threatens food security around the world is the emergence of insects capable of developing resistance to insecticides [1]. Excessive use of many of these insecticides is associated with various health and environmental issues [2,3,4]. *Liriomyza trifolii* is a highly polyphagous pest in crop fields and greenhouses that has detrimental economic impacts [5]. Both larvae and adults selectively eat only the layers with the least amount of plant cellulose [6]. Stippling is one example of the damage in crop plants caused by the sap-sucking female fly; internal mining caused by larvae is another such example. These various types of damage allow pathogenic fungi to enter the leaves through feeding holes. These types of damage also facilitate the mechanical transmission of some plant viruses [7,8]. Both leaf mining and leaf spotting can greatly reduce the level of photosynthesis in a plant [9], resulting in lower crop quality and yield. In this work, the study of the interactions between proteins and ligand inhibitors is proposed as for potential defense for increased resistance in crop plants [10]. The biological role of PIs is based on the inhibition of the proteins present in the guts of insects, which in turn reduces the availability of amino acids necessary for their growth and development [11]. The harmful effects of synthetic insecticides on the environment and on human health have drawn the attention of researchers to develop safer alternatives. In recent years, many studies investigated the chemical composition of plants and the possibility of using their extracts as bioinsecticides. In particular, these studies focused on probing the possibility of inhibiting the action of a number of enzymes that are found in common pests. Our work is complementary to the experimental results obtained by Mashamaite et al. [12], which showed that some chemicals extracted from natural materials such as plants can be effective compounds as biopesticides. Their work also suggested that compounds such as the (disodium 5′-inosinate and petunidin 3-glucoside), (calcium 5′-guanylate and chlorogenic acid), chlorogenic acid alone, (kaempferol-3,7-di-O-glucoside with hyperoside and delphinidin 3-glucoside), and (myricetin 3′-glucoside and hyperoside), can be used for the development of multitarget bioinsecticides. To confirm and reproduce the results obtained by the work of Mostafa et al. [13], it is necessary to perform structural modeling, binding site interaction prediction, molecular docking free energy calculations, binding pose analysis, dynamic stability and conformational perturbation analyses, radius of gyration analysis, hydrogen bond analysis, and molecular mechanics PBSA free energy calculations. The goal is to confirm, computationally, that these compounds could exert their bioactivities by altering the activities of acetylcholinesterase receptors, actin, α-tubulin, arginine kinase, and histone receptor III subtypes. Reproducing these results would also confirm the importance and versatility of the computational methods employed in this work in studying protein-ligand interactions.

We herein report the results of our structural modeling, binding site interaction prediction, molecular docking free energy calculations, binding pose analysis, dynamic stability and conformational perturbation analyses, radius of gyration analysis, hydrogen bond analysis, and molecular mechanics PBSA free energy calculations.

## 2. Results and Discussion

### 2.1. Database Search, Structural Modeling, and Model Validation

The homology modeling search of the query proteins sequences with the target *Liriomyza trifolii* proteins, namely acetylcholinesterase, α-tubulin, actin, arginine kinase, histone subunit III, Hsp90, and elongation factor 1-alpha, was performed using Blast on the NCBI server. The query coverage of proteins sequences showed (96%, 100%, 100%, 98%, 100%, 31% and 97%) with (54%, 61%, 60%, 58%, 59%, 56% and 55%) identity with the template proteins (1dx4.1.A, 5kx5.1.C, 4cbu.1.A, 4bg4.1.A, 4zux.1.A, 4cwr.1.A., and 5o8w.1.A), respectively. These were used as template proteins for the homology modeling of our target proteins. The Swissmodel server (https://Swissmodel.expasy.org/, accessed on 10 August 2022) generated (25, 35, 56, 58, 30, and 31) predictive models for *Liriomyza trifolii* proteins (acetylcholinesterase, *α-tubulin*, actin, arginine kinase, histone subunit III Hsp90, and elongation factor 1-alpha) with identity and Qualitative Model Energy Analysis (QMEAN) score values [14]. The models with low values of QMEAN scores were selected as the final models for in silico characterization and docking studies.

### 2.2. Structural Modeling, In Silico Characterization, and Model Validation

The selected models were verified for their stereochemical quality assessment. Furthermore, in each case of qualitative assessment, a comparative study was done with experimentally solved crystal structures to check the quality, reliability, accuracy, stability, and compatibility of the computationally predicted protein through a Ramachandran plot, the ERRAT score which is a so-called “overall quality factor” for nonbonded atomic interactions, with higher scores indicating higher quality [15], and the QMEAN score. The Ramachandran plot obtained through the PROCHECK module of the PDBSum server justified the stereochemical suitability of the predicted proteins. Acetylcholinesterase, α-tubulin, actin, arginine kinase, histone subunit III Hsp90, and elongation factor 1-alpha had 92.3 %, 94.3%, 94.6%, 94.2%, 96.4%, 93.5%, and 94.0% residues, respectively, accommodating in the most favored regions (A, B, and L). They also only had 7.7 %, 5.5%, 5.4%, 5.0%, 0.3%, 0.6%, and 5.2% residues occupied in the additionally allowed regions (a, b, l, and p), respectively (Table 1, Appendix A). Residues in generously allowed regions (a, b, l, and p) are (0.0%, 0.3%, 0.0%, 0.8%, 0.0%, 0.0%, and 0.4%) and residues in disallowed regions are (0.0%, 0.0%, 0.0%, 0.0%, 0.0%, 0.5% and 0.4%), respectively. The ERRAT scores for the modeled structure were found to be (100%, 89.3519%, 95.9596%, 93.1298%, 100%, 96.4824%, 92.5373%, and 81.7204%), respectively. The QMEAN score values of the models were (−0.11, −1.00, 0.13, 0.43, 0.25, 0.28, and 0.66), respectively. The three parameters suggested that the predicted model had satisfactory stereochemical quality and was close to the template structure.

### 2.3. Binding Site Prediction and Protein-Ligand Interaction

The putative ligand binding sites (both major and minor) for the predicted proteins were identified through Discovery studio software and were visualized (Figure 1). All target proteins (acetylcholinesterase, *α*-tubulin, actin, arginine kinase, histone subunit III, Hsp90, elongation factor 1-alpha, and carbomoylphosphate synthase) were docked with the ligands, most of which were phytochemicals derived from the leaves of *Phaseolus vulgaris* [16,17] and the yeast extract. We evaluated the protein-ligand interaction through SAMSON software [18]. It was found that the tool has discrepancies in results for accurate pose prediction among the various putative docking poses. 

### 2.4. Molecular Docking and Binding Free Energy Calculation

The prepared protein structures of (acetylcholinesterase, *α-tubulin*, actin, arginine kinase, histone subunit III, Hsp90, elongation factor 1-alpha, and carbamoyl phosphate synthase) were docked using SAMSON software with phytochemical compounds and yeast extracted compounds listed in the Appendix A. The results of the docking studies were provided in Table 2, and it was revealed that the phytochemical compounds were superior to the yeast extract compounds based on the docking score. All docking results were monitored by scoring functions that predict how well the ligand binds in a particular docked pose. This scoring function gives the ranking of the ligands. In the present study, the docking score was taken into consideration for the selection of the best ligands. This allowed us to explain the mechanisms of insect death. A mathematical empirical scoring function was used to approximately predict the binding affinity between two molecules after they have been docked by approximating the ligand’s binding free energy [20]. It includes various force field interactions such as electrostatic and van der Waals contributions, which influence ligand binding. Subsequently, the docked structures were queried for binding free energy calculation. The results of binding free energy calculation were provided in Table 2. It was found that binding energy values supported the docking result well. Hesperidin, Naringin, and Rosmarinic acid have higher binding energies than other compounds. All of the other values contribute to the ΔG values which reflect the binding energy of the protein-ligand complex.

### 2.5. Binding Pose Analysis

The binding mode of the compounds with proteins (acetylcholinesterase, α-tubulin, actin, arginine kinase, histone subunit III, Hsp90, elongation factor 1-alpha, and carbamoyl phosphate synthase) showed the different interactions between the proteins and ligands showed in Table 3. The interactions between the inhibitors and their target proteins, as well as their binding modes and orientations, are shown in Figure 2, Figure 3, Figure 4, Figure 5, Figure 6, Figure 7, Figure 8 and Figure 9.

#### 2.5.1. Root Mean Square Deviation (RMSD) Analysis

Calculations of the RMSD for the ligand-enzymes complex were used to determine the dynamic stability and conformational perturbation, which occur in each of the simulated systems during the simulation time scale. The RMSD values were calculated for the following protein-inhibitors combinations: acetylcholinesterase with disodium 5′-inosinate and petunidin 3-glucoside; actin with calcium 5′-guanylate D and chlorogenic acid; α-tubulin with chlorogenic acid alone; arginine kinase with kaempferol-3,7-di-O-glucoside I, hyperoside D, delphinidin 3-glucoside ID, and histone subunit III complexes with myricetin 3′-glucoside ID and hyperoside D. All the trajectories reached equilibrium state after 20 ns, as shown in Figure 10. The RMSD values for all complexes are observed to be stable during the 50 ns simulation.

#### 2.5.2. Radius of Gyration (Rg) Analysis

The Rg factor is best described for the stability of receptor-ligand complexes during the molecular dynamics simulations. The results demonstrate that the Rg values during different time points for the acetylcholine esterase, actin, α-tubulin, arginine kinase, and histone subunit III complexes to their respective ligands are constant during 50 ns simulation, which indicates the compactness of all of the proteins (Figure 11). 

#### 2.5.3. Hydrogen Bond Analysis 

The number of hydrogen bonds for the ligand-enzymes complexes are plotted over a 50-ns MD simulation interval (Figure 12). Since hydrogen bonds constitute a transient connection that provides stability to the receptor-ligand complex, they constitute an important factor to consider when discussing receptor-ligand stability. These bonds determine the specificity of the binding mode. In this study, we have calculated all of the hydrogen bonds for all of the complexes. The numbers of hydrogen bonds at different time points have been calculated, as shown in Figure 12. The average number of hydrogen bonds calculated for inhibitors (disodium 5′-inosinate and petunidin 3-glucoside), (calcium 5′-guanylate D and chlorogenic acid), chlorogenic acid, (kaempferol-3,7-di-O-glucoside I, hyperoside D, delphinidin 3-glucoside ID) and (myricetin 3′-glucoside ID, hyperoside D) are (0–6, 0–5), (0–7, 0–9), 0–9, (0–8, 0–9, 0–10), respectively. All of the predicated ligands have shown continuous hydrogen bonding during the 50 ns simulation, which demonstrates the stability of the complexes. The only exception was chlorogenic acid, which only shows stable hydrogen bonding in the span of 35 ns. 

#### 2.5.4. Root Mean Square Fluctuation Analysis (RMSF)

The RMSF value refers to the flexibility and mobility of structure—a higher value of RMSF indicates a loosely bonded structure with twists, curves, and coils, while a lower value of RMSF indicates a stable secondary structure, including α-helix and beta-sheets. Our RMSF analysis demonstrates that all of the ligands showed less conformational variations during binding and can act as stable complexes (Figure 13). 

#### 2.5.5. Molecular Mechanics Poisson-Boltzmann Surface Area Free Energy Calculations

The binding capacity of the ligand towards the receptor is quantitatively estimated by binding free energy analysis. Binding free energy is the summation of all non-bonded interaction energies. The binding free energy of the interactions between acetylcholine esterase, actin, α-tubulin, arginine kinase, and histone subunit III and the docked ligands has been estimated using the molecular mechanics Poisson-Boltzmann surface area tool (G_MMPBSA) [21]. This useful tool allows for efficient and reliable free energy simulation to model protein-ligand interactions. Our binding energy analysis spanning 50 ns MD simulation trajectories show that all ligands have a binding affinity towards enzyme inhibition and form stable complexes. Other different kinds of interaction energies, such as van der Waals energy, electrostatic energy, polar solvation energy, and solvent accessible surface area (SASA) energy, have been also calculated for all the Tools Shapes complexes (Figure 14). Results indicate that van der Waals, electrostatic, and SASA energy negatively contribute to the total interaction energy, while only polar solvation energy positively contributes to the total free binding energy. In particular, the contribution of van der Waals interactions is much greater than that of the electrostatic interactions in all cases except the complexes arginine kinase-delphinidin 3-glucoside and histone subunit-myricetin 3′-glucoside. Furthermore, the contribution of SASA energy is relatively small when compared to the total binding energy. The negative value of van der Waals energy also points to the significant hydrophobic interaction between the ligands and the enzymes [22].

#### 2.5.6. Principal Component Analysis (PCA)

Principal component analysis is a method that utilizes linear combinations of measured variables, which allows for the reduction of the dimensionality of data and helps identify the principal sources of variation. In molecular dynamics simulations, PCA is a popular method to account for the essential dynamics of the system on a low-dimensional free energy landscape [23]. To analyze the collective motion of all complexes, PCA analysis based on C-a atoms has been performed. It was observed that the first few eigenvectors of the principal components (PCs) of the structures play an important role and describe the overall motions of the entire system. These data suggest that kaempferol-3,7-di-O-glucoside ID has formed very stable complexes with arginine kinase and myricetin 3′-glucoside ID with histone subunit III, which can be considered as a lead compound (Figure 15).

Since it was previously found that the first five eigenvectors constitute the majority portion of the total dynamics of the whole system, we plotted only the first two eigenvectors against each other, where each dot represents correlated motions (Figure 16). The well-stable clustered dots signify the more stable structure, and low-clustered dots represent the weaker stable structure. 

## 3. Materials and Methods

### 3.1. Database Search, Structural Modeling, and Model Validation

All protein sequences were obtained from NCBI (https://www.ncbi.nlm.nih.gov/, accessed on 10 August 2022) in FASTA format and are mentioned by their Gen Bank accession number in Table 2. The *Liriomyza trifolii* NCBI taxonomy (tax ID: 32264) proteins were selected by searching all of the sequential homolog and orthologs using NCBI Blast server [24] with the default values, and against the nonredundant protein sequences. The sequences were retrieved in the FASTA format as an amino-acid sequence. The initial atomic structures of the proteins, based on homology modeling, were built using the Swissmodel server (https://Swissmodel.expasy.org/, accessed on 10 August 2022). In this study, a sequence of Blast-P similarities for recognition of closely related structural homologs in *Liriomyza trifolii* was queried against a PDB database [18]. The first hit on the annotation Blast-p was obtained to identify the templates based on PDB template ID. The Protein Data Bank collected the PDB file of the templates and was aligned using BLAST. The Swissmodel server used the target sequence file, the alignment file, the PDB file for the prototype, and all the template proteins to build the homology model. Homology models with a score of <−4 were chosen. The optimized models (acetylcholinesterase, α-tubulin, actin, arginine kinase, histone subunit III, heat shock protein 90 (Hsp90), and elongation factor 1-alpha) were found to be suitable based on several qualitative background checks, including the PROCHECK (PDBSum) and Swissmodel server (https://saves.mbi.ucla.edu/, accessed on 10 August 2022). The Ramachandran plot evaluated that the predicted models were closer to the template with (99.1%, 92.6%, 86.7%, 84.4%, 88.6%, 88.4%) residues lying in the favored regions. The ERRAT score values of 99.1304, 89.7527, 96.4539, 82.0707, 96.5217, 90.9774, and QMEAN score indicated that the predicted models were reliable and satisfactory, as they are higher than the ideal values of the QMEAN score <−4, and ERRAT around 95% for a model with a satisfactory resolution [24]. 

### 3.2. Preparation of Proteins and Ligands

The sequences of the *Liriomyza trifolii* proteins (acetylcholinesterase, actin, α-tubulin, arginine kinase, elongation factor 1-alpha, Hsp90, and histone subunit III) with GenBank accession no. number (CAI30732.1, ARQ84036.1, ARQ84030.1, ARQ84038.1, ARQ84034.1, AGI19327.1, ARQ84032.1, ABL07756.1, respectively) were obtained from NCBI. The protein sequences were retrieved in the FASTA format. The 3-D structures of proteins were built using the Swissmodel server (https://Swissmodel.expasy.org/, accessed on 10 August 2022). Here, proteins were selected as target receptor proteins and were imported to the 3-D refine server to perform energy minimization for the six proteins (http://sysbio.rnet.missouri.edu/3Drefine/, accessed on 10 August 2022). During docking studies, all water molecules and ligands were removed, and hydrogen atoms were added to the target proteins. The docking system was built using SAMSON software 2020 (https://www.samson-connect.net/, accessed on 10 August 2022). The structures were prepared using the protein preparation wizard of the Autodock Vina extension of SAMSON 2020 software. The X, Y, and Z dimensions of the receptor grid, used for the blind docking of ligands to proteins, are reported in Table 3. The ligands were retrieved from the PubChem database in SDF format. Subsequently, each ligand was converted into MOL2 format using OpenBabel software (http://openbabel.org/wiki/Main_Page, accessed on 10 August 2022), followed by an energy minimization at pH 7.0 ± 2.0 in SAMSON software.

### 3.3. Binding Site Prediction and Protein-Ligand Docking 

Discovery studio software and SAMSON software were used for binding site prediction. SAMSON software uses Autodock Vina as an extension to maximize the accuracy of these predictions while minimizing computer run-time [25]. It uses the interaction energy between the protein and a simple van der Waals probe to locate energetically favorable binding sites. The program is based on quantum mechanics, and it predicts the potential affinity, molecular structure, geometry optimization of the structure, vibration frequencies of coordinates of atoms, bond length, and bond angle. Following an exhaustive search, 100 poses were analyzed, and the best scoring poses were used to calculate the binding affinity of the ligands. The ligands that tightly bind to a target protein with high scores were selected in Table 3. The proteins were docked against a variety of bioactive compounds that are phytochemical components from the HPLC of leaves of *Phaseolus vulgaris* (ref) and yeast extract using SAMSON software [21]. The 2-D interaction was carried out to find favorable binding geometries of the ligand with the proteins using Discovery Studio software. Thus, the 2-D interaction images of the docked protein-ligand complexes with high scores to the predicted active sites were obtained. 

### 3.4. Protein Ligand Interaction Using SAMSON and Discovery Studio Software

The ligands were docked with the target proteins (acetylcholinesterase, actin, α-tubulin, arginine kinase, elongation factor 1-alpha, Hsp90, and histone subunit III), and the best docking poses were identified. Figure 1 and Figure 2 show the 2-D and 3-D structures of the binding poses of the compounds, respectively.

### 3.5. Protein–Protein Interaction Network

The *Liriomyza trifolii* proteins were submitted to the server for functional interaction associated network between partners for the STRING (Research Online of Interacting Genes/Proteins Data Basis version 10.0)13 [24]. The interactions were examined at medium and high confidences.

### 3.6. Molecular Dynamics Simulation

The molecular dynamic approach is widely used to assess atoms’ behavior and structural stability, and to study the conformational changes at an atomic level. Understanding the stability of protein upon ligand binding is significantly improved by molecular dynamics simulation studies. Gromacs 4.6.2 [26] with GROMOS96 54a7 force field [27] was used for MD simulation studies of two systems, at 50 ns each. The ProdrG2 Server was used to generate the topology of the analysis of enzyme-ligand complexes. Each system was placed in the center of the cubic box, with a distance of 1.0 nm between the enzyme and the edge of the simulation box. Each system was solvated with explicit water molecules. Before proceeding towards energy minimization, all systems were neutralized by adding Na^+^ and Cl^−^ ions, accordingly. The steepest descent method was used for the energy minimization of each system. MD simulations with NVT (isochoric-isothermal) and NPT (isobaric-isothermal) ensembles (N¼ constant particle number, V¼Volume, P¼Pressure, T¼Temperature) were performed for 1 ns, each, to equilibrate the enzyme-ligand system for constant volume, pressure (1 atm), and temperature (300 K). To calculate electrostatic interaction, the Particle Mesh Ewald (PME) algorithm [25] was used with a grid spacing of 1.6 Å and a cutoff of 10 Å, and the LINCS method was used to restrain the bond length. Finally, the trajectories were saved at every 2-fs time step and the production MD simulation of the enzyme-ligand complex was performed for 50 ns [28].

## 4. Conclusions

This study presented an array of naturally occurring, nontoxic, easily extractable, low-cost ligands that show great potential as inhibitors of a variety of proteins found in the gut of the polyphagous pest *L. trifolii* that is known to attack a myriad of crops. The target proteins are acetylcholinesterase, actin, α-tubulin, arginine kinase, and histone receptor III subtypes. The proposed inhibitors or inhibitor combinations are (disodium 5′-inosinate and praliciguat), (calcium 5′-guanylate and chlorogenic acid), chlorogenic acid alone, (kaempferol-3,7-di-O-glucoside with hyperoside and delphinidin 3-glucoside), and (myricetin 3′-glucoside and hyperoside), respectively. In lieu of an experimentally available structure of the target proteins, the initial models of the protein of *L. trifolii* origin were constructed using homology modeling. The analyses used in this investigation included structural modeling, binding site interaction prediction, molecular docking free energy calculations, binding pose analysis, dynamic stability and conformational perturbation analysis, radius of gyration analysis, hydrogen bond analysis, and molecular mechanics PBSA free energy calculations. The results demonstrated that the proposed inhibitors formed stable complexes with their target proteins while also having great potentials for inhibitory activity. All five ligand-protein complexes have favorable parameter values in RMSD, RMSF, RoG, intermolecular hydrogen bonding, and binding free energy for 50 ns. Trajectories analysis showed that the studied complexes displayed structural stability during the MD runs.

The are many various methods of predicting protein 3-D structures, for example, I-Tasser to obtain their ‘starting’ structures or AlphaFold server and Swissmodel server. Though the principle is the same for all of the homology modeling software, it is based on the template structure that the final model is built. Different software uses different templates to model, but we can conclude that the most exact commonly used online tool is Swissmodel; it is easy and the most widely feasible, and not too expensive to be used for predicting protein 3-D structures. Moreover, all of the various methods of predicting protein 3-D structures “yield the same predicted protein structures”.

The development of computer systems in biological studies has had a great impact on developing and understanding the effects of protein inhibitors. This allows the opportunity for optimizing and utilizing computational methods, such as the ones used in this study, as low-cost, efficient, and effective means of predicting protein-ligand interactions. 

## Figures and Tables

**Figure 1 ijms-23-12791-f001:**
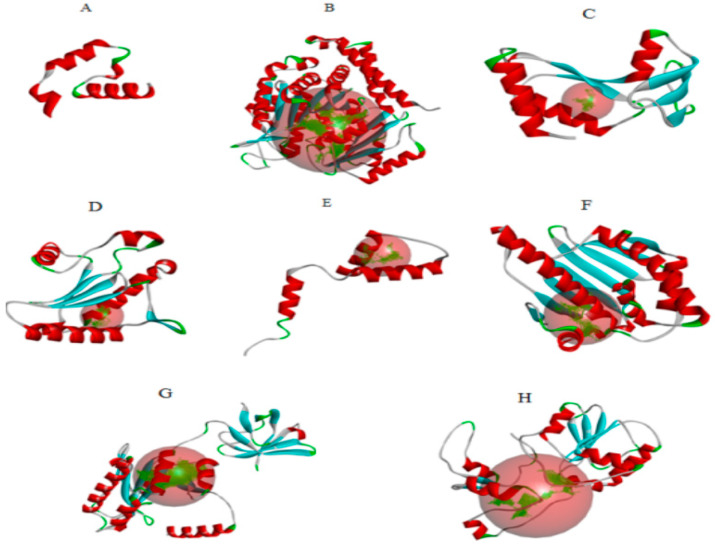
(**A**) acetylcholinesterase, (**B**) α-tubulin, (**C**) actin, (**D**) arginine kinase, (**E**) histone subunit III (**F**) Hsp90, (**G**) elongation factor 1-alpha, and (**H**) carbamoyl phosphate synthase of the *Liriomyza trifolii* modeled proteins through homology modeling using the Swissmodel server and visualized through the Discovery Studio 3.0 visualization tool [19]. The large red sphere represents the cavities surrounding the active sites and was visualized using the visualization module of Discovery Studio 3.0 visualization.

**Figure 2 ijms-23-12791-f002:**
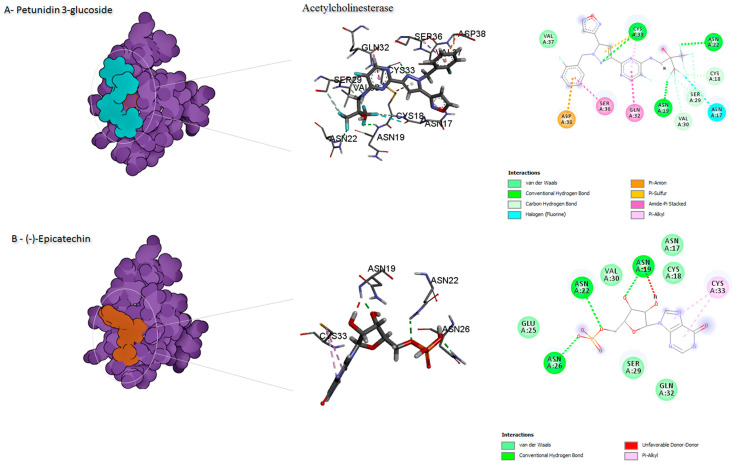
Predicted binding mode for acetylcholinesterase with petunidin 3-glucoside and disodium 5′-inosinate.

**Figure 3 ijms-23-12791-f003:**
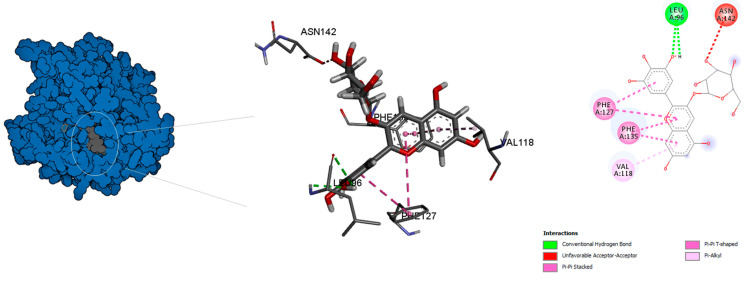
Predicted binding mode for α-tubulin with chlorogenic acid.

**Figure 4 ijms-23-12791-f004:**
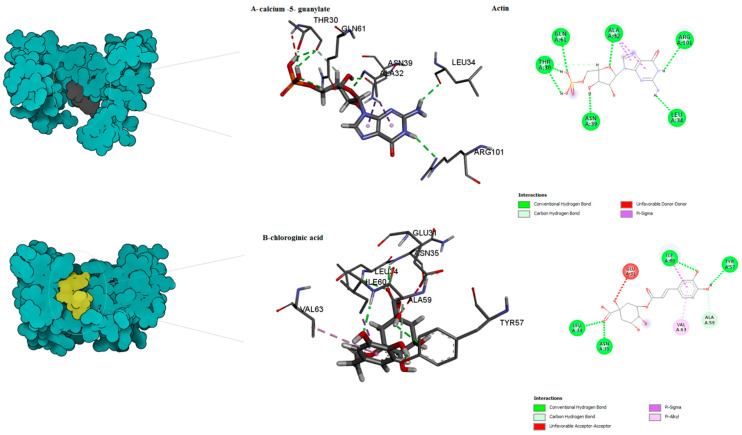
Predicted binding mode for actin with calcium 5- guanylate and chlorogenic acid.

**Figure 5 ijms-23-12791-f005:**
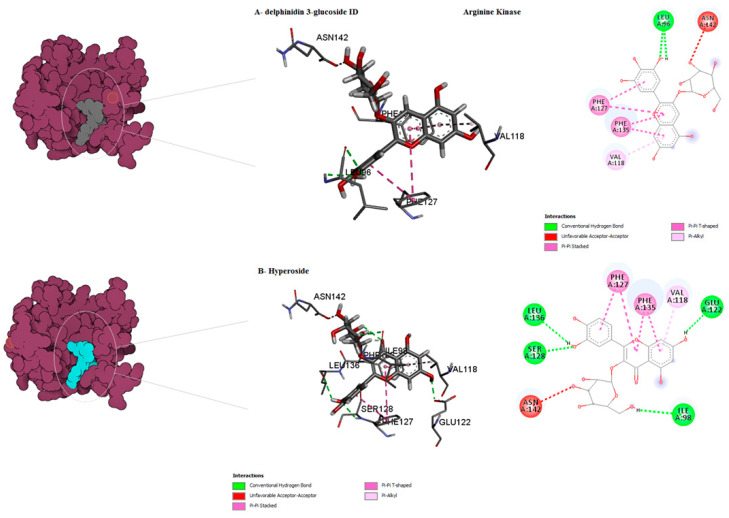
Predicted binding mode for arginine kinase with hyperoside and delphinidin 3-glucoside ID.

**Figure 6 ijms-23-12791-f006:**
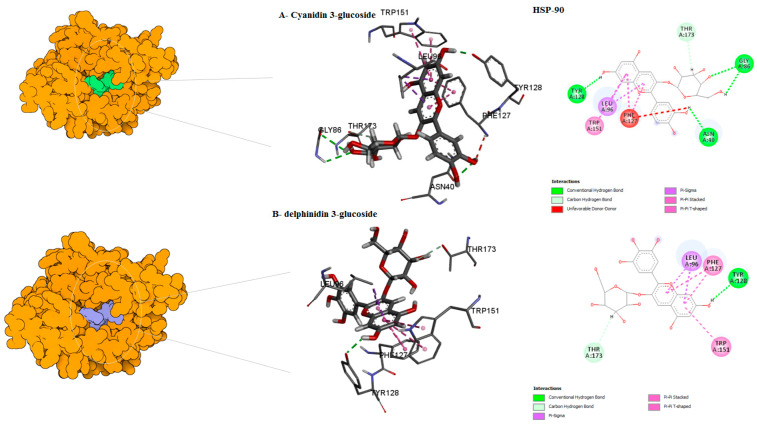
Predicted binding mode for Hsp90 with cyanidin 3-glucoside ID–1 and delphinidin 3-glucoside D/ID.

**Figure 7 ijms-23-12791-f007:**
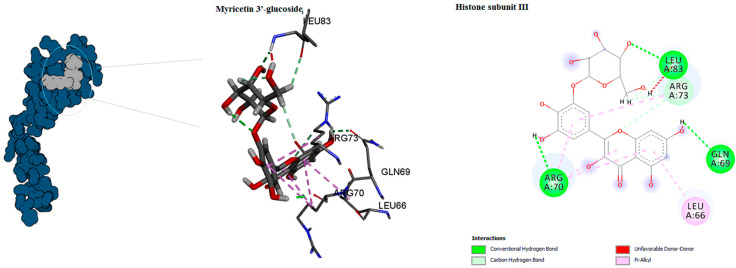
Predicted binding mode for histone subunit III with myricetin 3′-glucoside ID.

**Figure 8 ijms-23-12791-f008:**
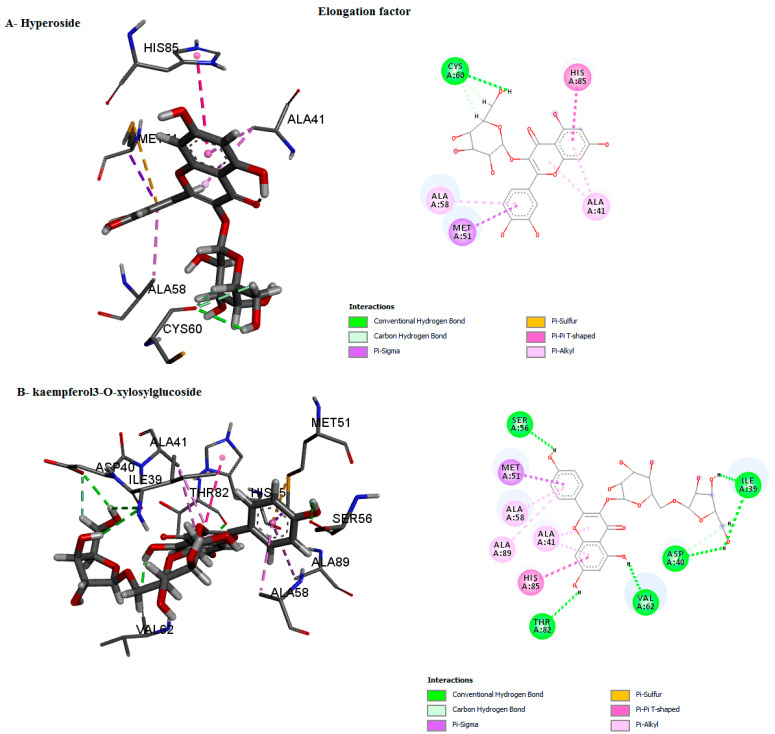
Predicted binding mode for elongation factor with kaempferol3-O-xylosylglucoside and hyperoside D.

**Figure 9 ijms-23-12791-f009:**
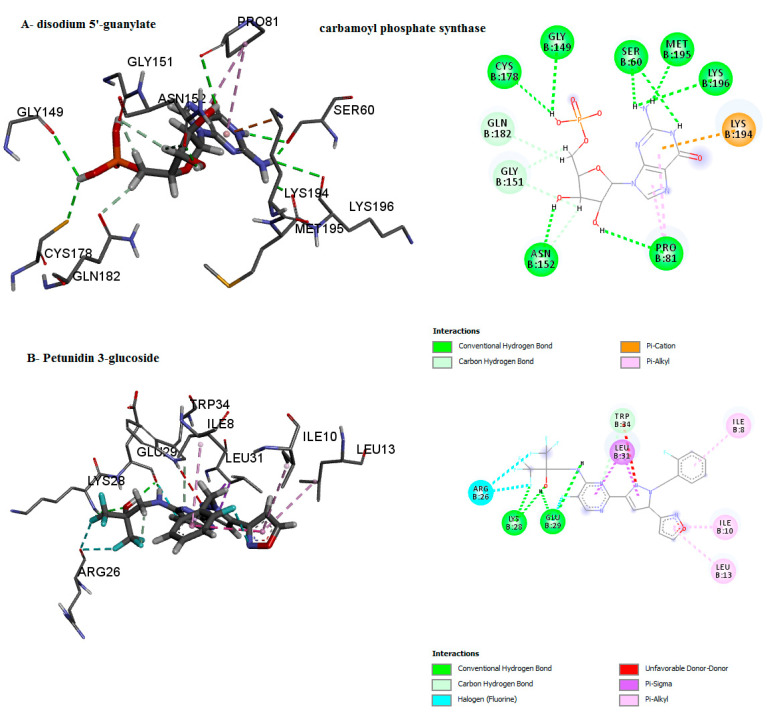
Predicted binding mode for carbamoyl phosphate synthase with disodium 5′-guanylate and petunidin 3-glucoside.

**Figure 10 ijms-23-12791-f010:**
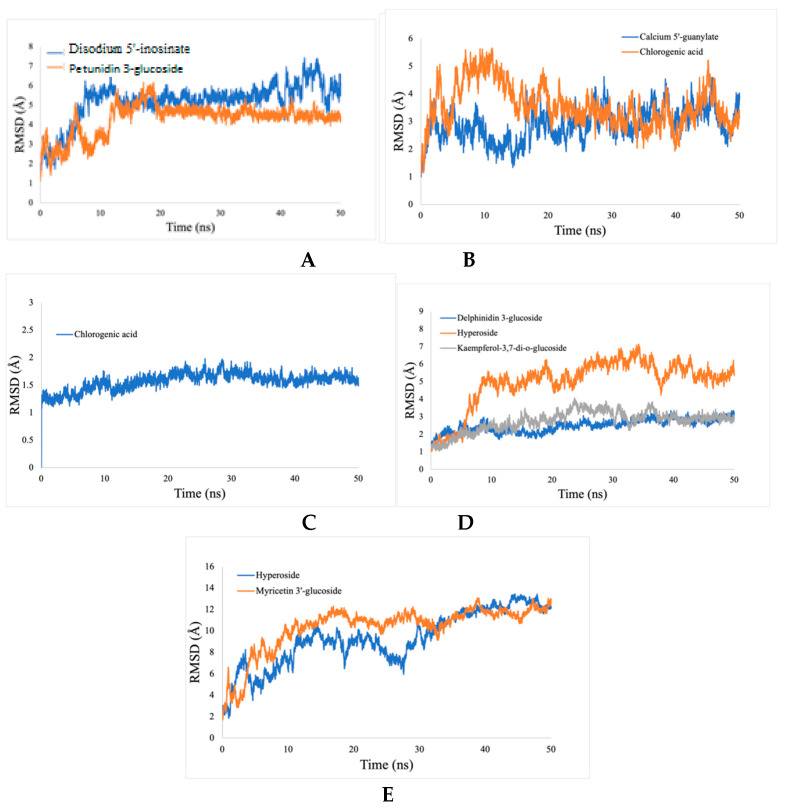
RMSD analyses of protein-ligand complexes. (**A**) acetylcholine esterase, (**B**) actin, (**C**) α-tubulin, (**D**) arginine kinase, and (**E**) histone subunit III with inhibitors (disodium 5′-inosinate and petunidin 3-glucoside), (calcium 5′-guanylate D and chlorogenic acid), chlorogenic acid, (kaempferol-3,7-di-O-glucoside I, hyperoside D, delphinidin 3-glucoside ID) and (myricetin 3′-glucoside ID, hyperoside D), respectively.

**Figure 11 ijms-23-12791-f011:**
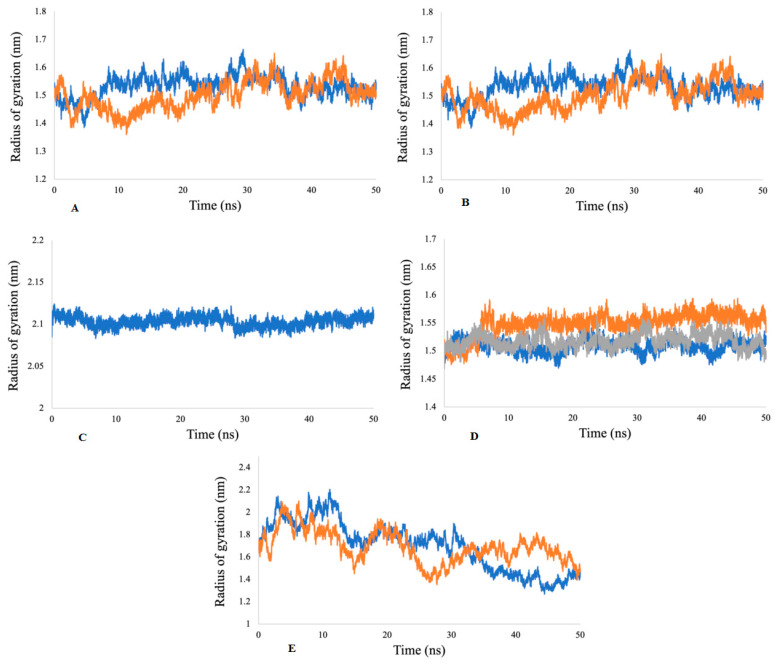
Radii of gyration for (**A**) acetylcholine esterase, (**B**) actin, (**C**) α-tubulin, (**D**) arginine kinase, and (**E**) histone subunit III with inhibitors (disodium 5′-inosinate and petunidin 3-glucoside), (calcium 5′-guanylate D and chlorogenic acid), chlorogenic acid, (kaempferol-3,7-di-O-glucoside I, hyperoside D, delphinidin 3-glucoside ID), and (myricetin 3′-glucoside ID, hyperoside D) complexes.

**Figure 12 ijms-23-12791-f012:**
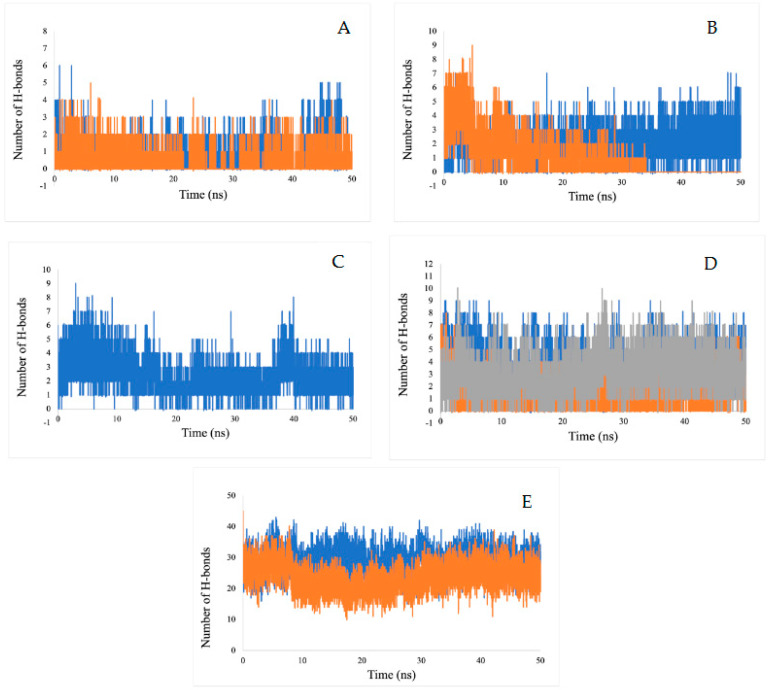
Estimation of the hydrogen bond number during a 50 ns MD simulation of for (**A**) actylcholine esterase, (**B**) actin, (**C**) α-tubulin, (**D**) arginine kinase, and (**E**) histone subunit III with inhibitors (disodium 5′-inosinate and petunidin 3-glucoside), (calcium 5′-guanylate D and chlgenic acid), chlorogenic acid, (kaempferol-3,7-di-O-glucoside I, hyperoside D, delphinidin 3-glucoside ID), and (myricetin 3′-glucoside ID, hyperoside D) complexes.

**Figure 13 ijms-23-12791-f013:**
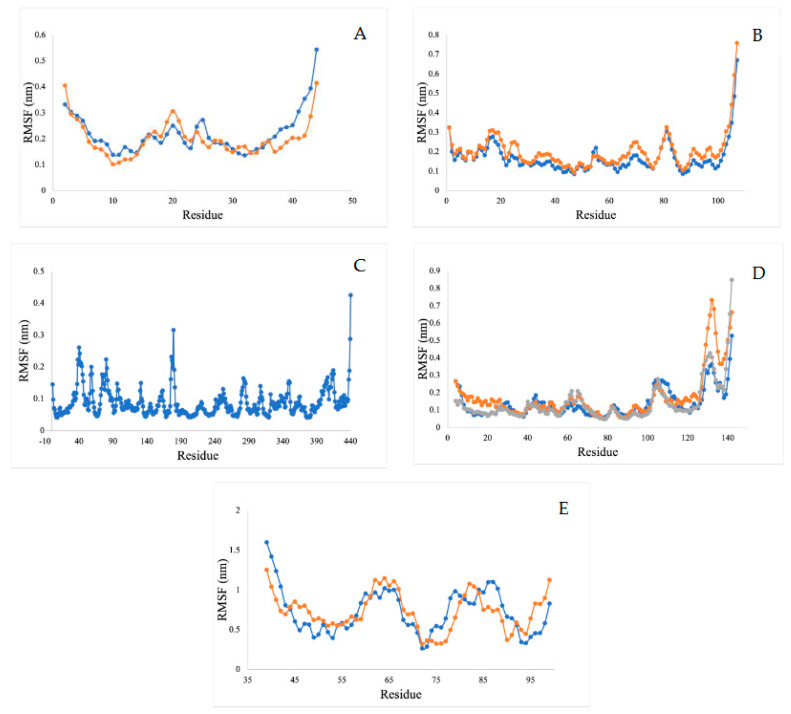
RMSF analysis for (**A**) acetylcholine esterase, (**B**) actin, (**C**) A-tubulin, (**D**) arginine kinase, and (**E**) histone subunit III with inhibitors (disodium 5′-inosinate and petunidin 3-glucoside), (calcium 5′-guanylate D and chlorogenic acid), chlorogenic acid, (kaempferol-3,7-di-o-glucoside I, hyperoside D, delphinidin 3-glucoside ID), and (myricetin 3′-glucoside ID, hyperoside D) complexes.

**Figure 14 ijms-23-12791-f014:**
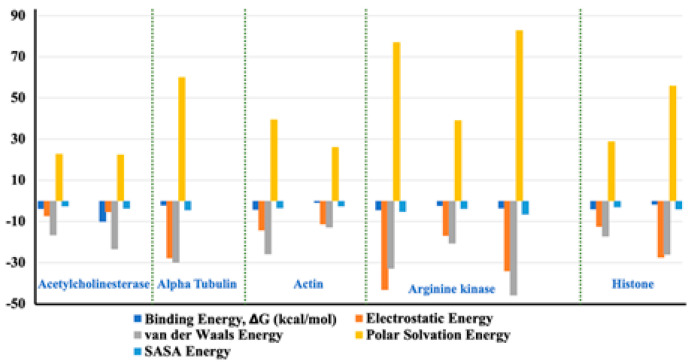
Representation of the van der Waals, electrostatic, polar solvation, SASA, and binding energy for docked compounds into: Acetylcholine esterase, Actin, A-tubulin, Arginine kinase and histone subunit III with inhibitors (disodium 5′-inosinate and petunidin 3-glucoside), (calcium 5′-guanylate D and chlorogenic acid), chlorogenic acid, (kaempferol-3,7-di-o-glucoside I, hyperoside D, delphinidin 3-glucoside ID), and (myricetin 3′-glucoside ID, hyperoside D) complexes.

**Figure 15 ijms-23-12791-f015:**
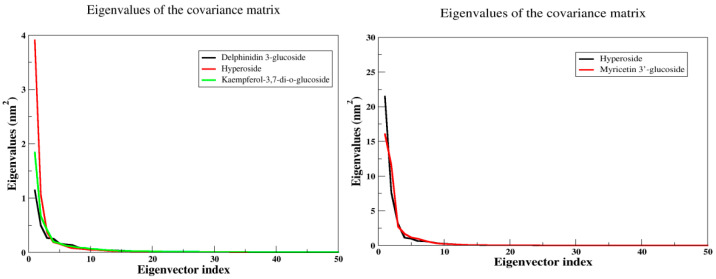
The PCA analysis, the plot of eigenvalues vs. eigenvectors have been considered.

**Figure 16 ijms-23-12791-f016:**
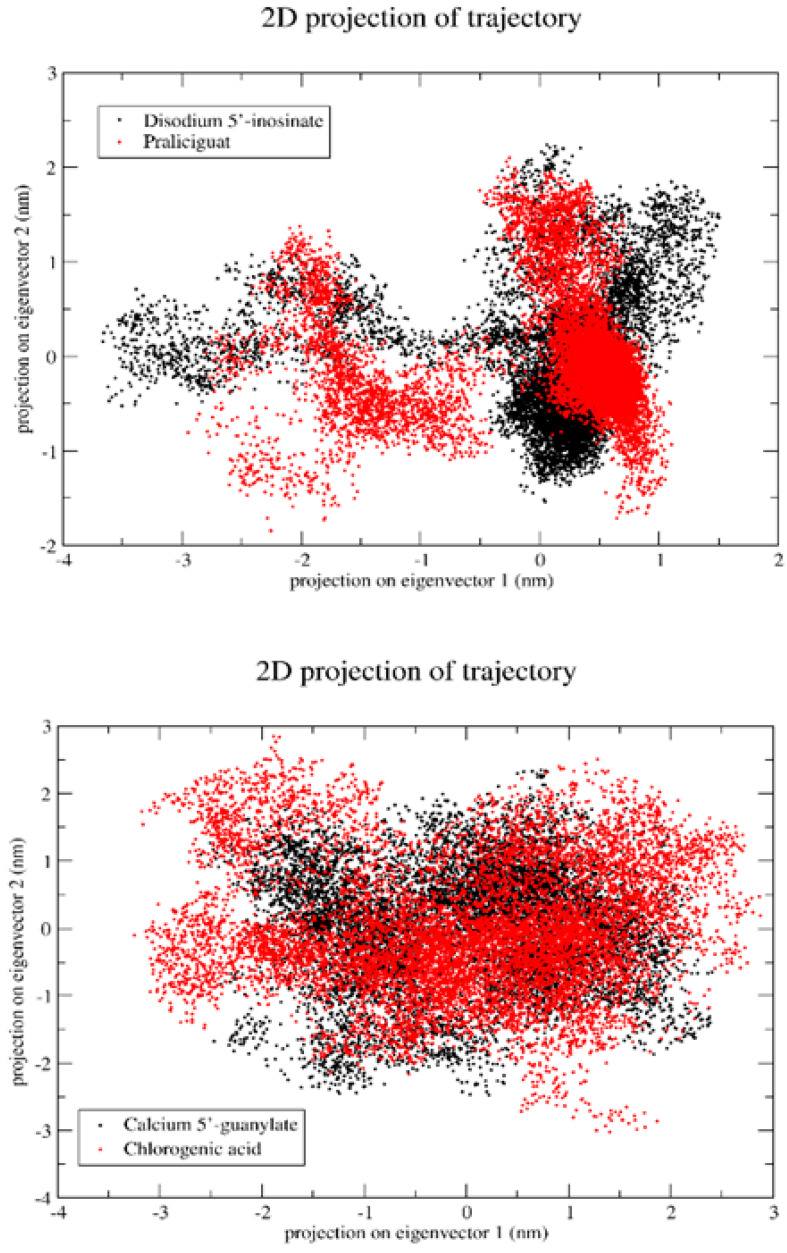
Projection of the motion of a protein in phase along the principal components (PC1 and PC2).

**Table 1 ijms-23-12791-t001:** Ramachandran plot values obtained through PROCHECK. Structurally and energetically favored regions are classified into allowed, generously allowed, and disallowed categories.

Protein	Ramachandran Plot Values			
	Core %	Allowed %	Generously %	Disallowed %
Acetylcholinesterase	92.3	7.7	0.0	0.0
α-tubulin	94.3	5.5	0.3	0.0
Actin	94.6	5.4	0.0	0.0
Arginine kinase	94.2	5.0	0.8	0.0
Histone subunit 3	96.4	0.3	0.0	0.0
Hsp90	93.5	0.6	0.0	0.5
Elongation factor	94.0	5.2	0.4	0.4
Carbamoyl phosphate synthase	85.6	13.2	0.0	1.1

**Table 2 ijms-23-12791-t002:** Protein Genbank sequences and modeling parameters for building with template ID, sequences identities, the coverage of the protein, and quality mean estimation for validation of the protein in quality.

Protein	Genbank:	Template	Seq Identity	Coverage	QMEAN
Acetylcholinesterase	CAI30732.1	1dx4.1.A	54%	96%	−0.11
α-tubulin	ARQ84036.1	5kx5.1.C	61%	100%	−1.00
Actin	ARQ84030.1	4cbu.1.A	60%	100%	0.13
Argenin Kinase	ARQ84038.1	4bg4.1.A	58%	98%	0.43
Histone Subunit3	ARQ84034.1	4zux.1.A	59%	100%	0.25
Hsp90	AGI19327.1	4cwr.1.A	56%	31%	0.28
Elongation Factor	ARQ84032.1	5o8w.1.A	55%	97%	0.66

**Table 3 ijms-23-12791-t003:** Docking scores and molecular properties of bioactive phytochemical components from the HPLC of *Phaseolus vulgaris* leaves (ref) and the yeast extract compounds. The molecular docking table shows the protein target, the interactive ligands with the highest binding energy, the type of bound of the highest score ligands, and the X, Y and Z geometry values of the protein.

Target	Ligands	Binding Energy of Direct Kcal/mole	Binding Energy of Indirect Kcal/mole	Binging Site	Type of Bond	X, Y and Z Value
Acetylcolenestras	Disodium 5′-inosinate	−6.5	0	SER36, CYS33, SER29, ASN22, ASN19, CYS18		center_x = 15.4462 center_y = 85.5487 center_z = −1.2087
	Petunidin 3-glucoside	0	−6.9	ASN17, CYS18 ASN19, ASN22 SER29, VAL30 GLN32, CYS33, SER36VAL37, ASP38	Alkyl Conventional H-bond carbon H-bond Amide -pi stacked pi-Alkyl Van der waals pi-Alkyl pi-Anion Halogen (Fluorine) Pi-Sulfur	
α-tubulin	Chlorogenic acid ID	0	−10.3	GLN11, ALA12 ASP69, ALA100 GLY144, ILE171 TYR224, ASN228	Conventional H-bond pi-Alkyl Amide-pi stacked Pi-Pi Stacked Pi-Sigma Unfavourable Donor–Donor Unfavourable Acceptor–Acceptor	center_x = 12.8911 center_y = 28.1478 center_z = −3.7346
Actin	Calcium 5′-guanylate D	−7.6	0	THR30, ALA32 LEU34, ASN39, GLN61, ARG101	Conventional H-bond cabon H-bond Unfavourable Donor–Donor Pi-Sigma	center_x = 16.2688 center_y = 30.1281 center_z = 30.8989
	Chlorogenic acid ID	0	−7.4	GLU31, LEU34 ASN35, TYR57 ALA59, ILE60 VAL63	Cabon H-bond Pi-Sigma pi-Alkyl Conventional H-bond Unfavourable Acceptor–Acceptor	
Arginine kinase	Kaempferol-3,7-di-O-glucoside I	0	−9.9	PHE135, SER128 PRO126, ILE98 HIS95	cabon H-bond Conventional H-bond Unfavourable Donor–Donor Pi-Pi Stacked	center_x = 21.1887 center_y = −3.9607 center_z = 13.2428
	Hyperoside D	−9	0	ASN142, LEU136, PHE135, SER128, PHE127, VAL118, ILE98	Van der waals Conventional H-bond Unfavourable Acceptor–Acceptor pi-Alkyl Pi-Pi Stacked Pi-Pi T–shaped	
	Delphinidin 3-glucoside ID	−9	−9	ASN142, PHE135, PHE127, VAL118, LEU96	Van der waals Conventional H-bond Unfavourable Acceptor–Acceptor pi-Alkyl Pi-Pi Stacked Pi-Pi T–shaped	
Hsp90	Cyanidin 3-glucoside ID –1	0	−10.9	ASN40 GLY86 LEU96 PHE127 TYR128 TRP151 THR173	cabon H-bond Conventional H-bond Unfavourable Donor–Donor Pi-Pi Stacked Pi-Pi T–shaped Pi-Sigma	center_x = 0.6349 center_y = 14.4620 center_z = 20.6177
	Delphinidin 3-glucoside D/ID	−10	12	ASN40, ASP82 GLY86, LEU96, GLY126 PHE127, TRP151 THR173	Cabon H-bond Conventional H-bond Pi-Sigma Pi-Pi T–shaped Pi-Pi Stacked	
	Hyperoside D	−9	0	ASN40 ASP82 GLY86 LEU96 GLY126 PHE127 TRP151 THR173	Conventional H-bond Pi-Sigma Pi-Pi T–shaped Pi-Pi Stacked	
Histone subunit3	Myricetin 3′-glucoside ID	0	−8.5	LEU66 GLN69 ARG70 ARG73 LEU83	Cabon H-bond Conventional H-bond Unfavourable Donor–Donor pi-Alkyl	center_x = 74.957 center_y = 39.323 center_z = −20.6626
	Hyperoside D	−6		TYR55 SER58 THR59 GLU60		
Elongation factor	kaempferol3-O-xylosylglucoside,	0	−9.8	HIS85, ALA58 GLN57, MET51 ALA41, ASP40, THR37	Van der waals Conventional H-bond pi-Alkyl Pi-Pi T–shaped Pi-Sigma Pi-Sigma	center_x = 22.1139 center_y = 30.3465 center_z = 31.8549
	Delphinidin 3-glucoside ID/D	−9.5	−9.5	HIS85, VAL62 CYS60, ALA58 GLN57, MET51 ALA41, THR37	Cabon H-bond Conventional H-bond Pi-Sigma Pi-Pi T–shaped pi-Alkyl Pi-Sulfur	
	hyperoside D	−9.9	0	HIS85, CYS60, ALA58, MET51, ALA41, THR37	Cabon H-bond Conventional H-bond Pi-Sigma Pi-Pi T–shaped pi-Alkyl Pi-Sulfur	
Carbamoyl phosphate synthase	disodium 5′-guanylate	−8.1	0	LYS196, MET195, LYS194, GLN182, CYS178, ASN152, GLY151, GLY149, PRO81, SER60	Cabon H-bond Conventional H-bond pi-Alkyl Pi-Cation	
	Petunidin 3-glucoside	0	−6.8	TRP34, LEU31, GLU29, LYS28, ARG26, LEU13, ILE10, ILE8, PHE2	Cabon H-bond Conventional H-bond Pi-Sigma pi-Alkyl Unfavourable Donor–Donor Halogen (Fluorine)	center_x = 21.3610 center_y = 59.0995 center_z = 103.8306

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
