# Peer review of "Nontoxic and Naturally Occurring Active Compounds as Potential Inhibitors of Biological Targets in *Liriomyza trifolii"

_ijms, 2022, doi:10.3390/ijms232112791_

Round 1

Reviewer 1 Report

The authors focused on using computational studies to find some protein inhibitors which will potentially work as an insecticide for Liriomyza trifolii. The research design for computational study is good. However, there are three major issues with this manuscript. (1) I did not see experimental validation for all the PIs they found promising. This caused a problem, that their research cannot be proven to be corrected. (2) The methods they used also lacked validation or evidence to prove their usefulness and correctness. They need at least a comparison or an example that how peers used these methods in a similar system, and how peers' results matched the experimental data. (3) The analysis of this paper is too general and sometimes lacks cretiria. We want to see a rational analysis by comparing your results with a known bar or an appropriately explained standard. 

Author Response

Dear sir, thank you for your comments, I hope to be our manuscript after revision acceptable to publication with regards 

Reviewer 2 Report

Page 1

The damage in crop plants is caused by the drip by the female fly ...

What does this mean. Drip?

Page 2

The term QMEAN is used seven times in the manuscript, first on page 2, but it is defined only on page 19, without a citation. 

It needs to be defined at the fist point of use and be cited there as well. 

Similarly, the terms ERRAT score, QMEN score on page 2 need to be explained and cited. 

The caption of Figure 1 has several problems. 

Here is the text in question.

Here, red region indicates favored region, yellow region for allowed and light yellow shows generously allowed region and white for disallowed region. Phi and Psi angels determine torsion angels. Based on an analysis of the structures of resolution of at least 2.0 Å and R-factor no greater than 20%, a good quality model would be expected to have over 90% in the most favoured regions

In these two sentences, there is a problem of English language usage

Here, red region indicates favored region, ....

should be

Here, the red color indicates a region that is favored energetically and structurally .... 

Phi and Psi angels?

I assume that the authors do not intend to refer to hypothetical religious entities.  Perhaps the authors mean Psi and Phi angles?

However, this sentence that contains these "angels" makes no sense to me. 

In this sentence,

Based on an analysis of the structures of resolution of at least 2.0 Å and R-factor no greater than 20%, a good quality model would be expected to have over 90%. 

What does this 90% refer to?  

Table 2 should explain the terms Core %, Allowed %, Generously %, and Disallowed %.

I have no idea what is meant here. 

Page 4:

All target proteins (acetylcholinesterase, α-tubulin, actin, arginine kinase, histone subunit III, Hsp90, elongation factor 1-alpha, and Carbomoylphosphate synthase) were docked with the ligands, which are phytochemical derived from the leaves of Phaseolus vulgaris [21, 22] and the yeast extract 

First, the above sentence should end with a period. Second, are all of the ligands used here phytochemicals that are derived from the leaves of Phaseolus vulgaris and the yeast extract? I know of one ligand used here that is a synthetic organic chemical. 

In the next sentence, 

We have evaluated the protein-ligand interaction through SAMSON software.

SAMSON needs a citation. 

The so-called SWISSMODLE server needs a citation and needs to be spelled correctly. This error is made in several parts of the manuscript. It should be Swissmodel.

The Discovery Studio 3.0 visualization tool. needs a citation and is incorrectly capitalized in the next sentence. 

Page 5

An empirical scoring function approximates the ligand binding free energy. This scoring function needs to be cited and explained. 

In the sentence

Hesperidin, Naringin, and Rosmarinic acid have higher binding energy than other compounds.

Since binding energies (this should be given in the plural) are generally given as negative delta G values, does this sentence indicate that Hesperidin, Naringin, and Rosmarinic acid are predicted to bind less well? Or does it mean that they bind better than others? Also, what do they bind to?

Page 12 and 13. 

Root mean square deviation (RMSD) analysis; Radius of gyration (Rg) analysis;  Hydrogen bond analysis

These analyses should be cited and explained. How were they done? What do the graphs on the next pages mean? How exactly do those graphs help lead to any conclusions?

The caption for Figure 13 suggests that these are molecular dynamics calculations (if that is the correct interpretation of the abbreviation MD in that figure caption.) 

Page 17 needs an explanation and citation for Principal component analysis (PCA).

Page 18 

Figure 17. projection of the motion of a protein in phase along the PC1 and PC2

The word projection needs to be capitalized, and PC1 and PC2 need to be explained. 

On page 19 Swissmodel is again and repeatedly, misspelled and mis-capitalized.  

Page 20 suggests that the compounds under study are natural and also nontoxic. If they are nontoxic, then the authors ought to cite the LD50 for these compounds. For instance, praliciguat, used here, is a synthetic compound that inhibits human adenylate cyclase, and is under investigation as a therapy for diabetic neuropathy. I cannot imagine that it is nontoxic at high doses.

It is odd that this activity towards a human enzyme is not mentioned in this manuscript. What about the other compounds used in this study. How strongly do they interact with other enzymes? These activities ought to be cited and compared to the values predicted in this paper. 

Lastly, is there a reason why not even one of these compounds or their analogs have not been tested experimentally? I understand that the insect enzymes may be hard to obtain, but there are, as the authors cite, homologous enzymes in other species. Doing so would be an important and even essential control for this work. 

Author Response

(The authors gave the same response as above.)

Round 2

Reviewer 1 Report

Agree to be published on IJMS. 

Author Response

Dear sir, thank you for your suggestion, I hope to our manuscript will be acceptable to publish after revision with warm regards 

Reviewer 2 Report

This is an interesting paper that uses advanced methods for protein structure prediction and methods to predict protein-ligand interactions. It lacks any controls. The language in the paper needs improvement. 

First, about the lack of controls in this paper:  It would be best to use experimental laboratory work to generate controls, laboratory work using the same ligands that are used here with the proteins used in this paper, proteins that are expressed in a suitable organism. It might be possible to use similar proteins that may be commercially available instead of using proteins generated using genetic technologies.

However, if the authors do not have access to those laboratory technologies, then I have a different suggestion for controls. 

This is to take existing protein-ligand structures from the pdb database, protein-ligand structures for which there are literature dissociation constants. These would be for one or more of the enzymes under study. They would not necessarily be structures of Liriomyza trifolii proteins, but of others that are related.

Then the peptide sequences of those structures can be processed in the exact same way as the Liriomyza trifolii sequences were processed in this paper, first to generate new predicted structures. The ligands for which there are experimental literature dissociation constants can be used in the same way, using the same computational methods, as is described for the ligands used in this paper. 

Then those control results, done in the exact same way as those done in this paper, can be compared to the experimental literature protein-ligand dissociation constants. The predicted control 3-D protein structures can also be compared with the experimental (NMR or X-Ray Crystallography) protein structures. To compare protein-ligand dynamics, something emphasized in this paper, it might be best to use multi-model pdb structures derived using NMR techniques. 

Another control is recommended:

The authors used the SwissModel server to predict protein structures. Did they also try the AlphaFold server, which has gotten so much publicity of late? Or the I-Tasser server? Do these various servers all yield the same structures?  Doing this is a necessary control. 

Here are some detailed comments about language. They are not exhaustive. The authors' text is in italics.

Page 2: 

Our study of the interactions between proteins and ligand inhibitors. 

is not a complete sentence

Our results complement the experimental results obtained during [??] naturally occurring active compounds as biopesticides.

What does this sentence mean; what does the word during in this sentence indicate?

Page 5

Hesperidin, Naringin, and Rosmarinic acid have higher binding energies than other compounds

Are the binding energies the same as the delta G values?

Table 3: 

Binding energy of Direct 

Binding energy of indirect

What do the words Direct and Indirect mean? What are the units of these binding energies? Kcal/mole perhaps?

On Pages 8-11 the ball and stick structures given for the ligands involved here only serve to hide the ligand structures. The spheres should be replaced by the letters, C, S, O, N, etc. denoting the relevant elements, and the lines connecting the atoms ought to reflect the number of bonds involved. 

The labels (such as Cys:178) adjacent to these structures are so fuzzy that they are almost impossible to read. These images ought to be redone. 

Page 14

Hydrogen bond for the ligand-enzymes complex are plotted along the 50-ns MD simulation. 

I do not understand this sentence. Does it mean the number of hydrogen bonds, the strength of hydrogen bonds, or something else? The text and figures that follow suggest that the authors mean the number of hydrogen bonds, but I cannot be sure.  The text on this page does not say how these hydrogen bond numbers were calculated. The same comment holds for these other types of results that follow. They are:

Root mean square deviation (RMSD) analysis 
Radius of gyration (Rg) analysis
Principal component analysis (PCA)

The text in this part of the paper should say what methods were used. 

In Figure 14, the authors should tell the reader what docked compounds were used in each case, even though this is described elsewhere.

Page 18

The well-stable clustered dots signify the more stable structure and low-clustered dots represent the weaker stable structure.

The associated figure has dots of different colors on different parts of the graph. How is the reader supposed to infer which dots are well-stable clustered dots and which dots are low-clustered dots? What do these words mean?

Author Response

(The authors gave the same response as above.)

Round 3

Reviewer 2 Report

I very much like this paper, but there are problems.

There continue to be problems in English language usage. I cannot address them all. As an example, consider the following sentences. 

Both larvae and adults selectively eat only the layers with the least amount of plant cellulose [6]. The damage in crop plants is caused by stippling resulting for the sap-sucking female fly as well as internal mining by larvae, which allows pathogenic fungi to enter the leaf through feeding holes and mechanical transmission of some plant viruses [7, 8]. 

I believe that it means the following, but I am not sure. 

Both larvae and adults selectively eat only the layers with the least amount of plant cellulose [6]. Stippling is one example of the damage in crop plants caused by the sap-sucking female fly; internal mining caused by larvae is another such example. These various types of damage allow pathogenic fungi to enter the leaves through feeding holes. These types of damage also facilitate the mechanical transmission of some plant viruses [7, 8]. 

Here is another example:

The biological role of PIs is based on the inhibition of proteins present in the guts of insects, which in turn, reduces the availability of amino acids necessary for their growth and development [11].

the inhibition of proteins ?? Does this not mean the inhibition of protein degradation or hydrolysis?

Autodocking Vina is more correctly written as Autodock Vina

Consider this sentence fragment.

.... it appears that this compounds could exert their bioactivity by modifying the activity of acetylcholinesterase receptors, actin, α-tubulin, arginine kinase, and histone receptor III subtypes, as was suggested and confirmed in previous experimental assays [12, 13]. 

not this compounds, but these compounds. 

Finally, the authors need to think about how to generate controls for this work.

Are these previous assays, cited in the above sentence, a basis for possible controls in the current work?  Could they compare the previous experimental assays with this work? Could they extend this work so as to facilitate those comparisons?

Could they apply their methods to related enzymes for which experimental 3D structures are available, and for which experimental dissociation constants are available? Can they compare the results found when using  different docking methods?

All these things would be essential controls. 

Author Response

Dear sir, thank you for your notes and comments, we do all the necessary changes and we hope for our manuscript to be acceptable to publishing with regards 

Round 4

Reviewer 2 Report

The authors have responded in a productive way to the most important comments made in previous reviews. While the authors have reacted to the request for controls, that reaction has not resulted in a change in the manuscript. For instance, in the authors' response, they write that 

We can, however, add the suggested control experiments as future work at the end of the paper to account for the reviewer’s honest and thorough critique. In addition, all these different servers yield the same predicted protein structures Database.

That addition, describing "future work at the end of the paper" is not visible to me in the version of the paper that I have available. Also, it would be simple and easy to state that the various methods of predicting protein 3-D structures "yield the same predicted protein structures." But I cannot see where they actually state that in the manuscript. 

Author Response

We thank you for your interest and comments. We have worked to add what you requested. We hope now that our manuscript will be suitable for publication.
